# Amelioration of Full-Thickness Wound Using Hesperidin Loaded Dendrimer-Based Hydrogel Bandages

**DOI:** 10.3390/bios12070462

**Published:** 2022-06-27

**Authors:** Praveen Gupta, Afsana Sheikh, Mohammed A. S. Abourehab, Prashant Kesharwani

**Affiliations:** 1Department of Pharmaceutics, School of Pharmaceutical Education and Research, Jamia Hamdard, New Delhi 110062, India; guptapraveen915@gmail.com (P.G.); afsana.pharmacist@gmail.com (A.S.); 2Department of Pharmaceutics, College of Pharmacy, Umm Al-Qura University, Makkah 21955, Saudi Arabia; maabourehab@uqu.edu.sa; 3Department of Pharmaceutics and Industrial Pharmacy, College of Pharmacy, Minia University, Minia 61519, Egypt

**Keywords:** wound healing, dendrimer, hydrogel, bandage, full-thickness wound, nanotechnology, anti-bacterial, hesperidin

## Abstract

Wound healing is a complex biological phenomenon, having different but overlapping stages to obtained complete re-epithelization. The aim of the current study was to develop a dendrimer-based hydrogel bandage, to ameliorate full-thickness wounds. Hesperidin, a bioflavonoid found in vegetables and citrus fruits, is used for treatment of wounds; however, its therapeutic use is limited, due to poor water solubility and poor bioavailability. This issue was overcome by incorporating hesperidin in the inner core of a dendrimer. Hence, a dendrimer-based hydrogel bandage was prepared, and the wound healing activity was determined. A hemolysis study indicated that the hesperidin-loaded dendrimer was biocompatible and can be used for wound healing. The therapeutic efficacy of the prepared formulation was evaluated on a full-thickness wound, using an animal model. H&E staining of the control group showed degenerated neutrophils and eosinophils, while 10% of the formulation showed wound closure, formation of the epidermal layer, and remodeling. The MT staining of the 10% formulation showed better collagen synthesis compared to the control group. In vivo results showed that the preparation had better wound contraction activity compared to the control group; after 14 days, the control group had 79 ± 1.41, while the 10% of formulation had 98.9 ± 0.42. In a nutshell, Hsp-P-Hyd 10% showed the best overall performance in amelioration of full-thickness wounds.

## 1. Introduction

Wounds represent the physical injuries that occur when the skin is broken externally or internally, which causes the opening or fracturing of skin. Various factors, such as temperature, oxygenation, and infections, as well as some other factors such as age, asthma, sex hormones, auto-immune disorders, smoking, obesity, etc., can interfere in wound healing [1]. It is a natural phenomenon that restores the integrity of the skin and also mitigates skin injury [2]. Four basic phases are involved in the treatment of a wound: hemostasis, the inflammatory phase, proliferation phase, and remodeling phase. Different cells, components of the ECM (extracellular matrix), mediators such as inflammatory cytokines, and different growth factors are involved in the wound healing process. Migration and proliferation of fibroblasts and re-epithelization by keratinocytes are essential in reducing the area of the wound [3]. In the case of different types of wounds, an acute wound, severe skin injuries, second and third-degree burns, and leg ulcers may not result in complete restoration of the function of tissues [4]. In the US alone, approximately over 6.5 million people suffer from wound injuries [5]. It has been estimated that globally wound care expenses will reach more than 24.8 billion dollars by 2024 [6]. Therefore, complete wound healing is essential, to overcome the cost of wound healing treatments and inhibit the progress of the chronic wound. Many researchers have focused on developing an efficient method to hasten the wound healing process [7].

Nanotechnology has emerged as an important field, due to the distinctive features of nanoparticles [8]. They have sizes in the range of 10–1000 nm [9]. Nanoparticles possess biological properties that are usually based on their physical and chemical properties [10]. Recently, polymers have gained much attention due to various properties, such as their safety, economical use, and being environmentally friendly [11]. In order to overcome the various challenges of traditional delivery vectors, scientists have focused more on nanoparticles, due to their adequate stability, safety, high surface to volume ratio, and cost-effective range [12]. Dendrimers are highly symmetric, spherical, and highly branched, with a nano-range 3D well-defined structure, size, and surface charge [13]. Due to their chemical homogeneity, the chance of increasing their size by repeated addition of a chemical group and their high-density of surface groups make them a suitable candidate for various biomedical applications [14]. Dendrimers such as Poly(amidoamine), also called PAMAM, are among the family of mono-disperse and highly branched units with a well-defined structure. The cationic primary amine group at their surface allows them to participate in binding with other chemical entities [15]. PAMAM dendrimers are mostly used as a vector for the delivery of various hydrophilic or hydrophobic drugs and genes [16]. Apart from this, dendrimers are a good choice because of their high drug payload, passive targeting, solubilization, increase in half-life loaded drug, purity and stability, globular shape, and monodispersity [17].

Recently, many natural source-derived plant-based compounds have been used in the treatment of various diseases [18]. Hesperidin is a bioflavonoid that can be found in vegetables and citrus fruits such as oranges or lemons [19]. Its aglycone portion is known as hesperetin, which represents the non-sugar portion of flavonoids [20]. The peels of citrus fruits have the highest concentration of hesperidin [21]. Hesperidin has shown a variety of pharmacological properties, such as being anti-microbial, anti-oxidant, anti-inflammatory, and analgesic, etc. [22]. Topical application of hesperidin gel has shown effective results in the treatment of wounds. However, due to its poor solubility and bioavailability, hesperidin failed to deliver the desired therapeutic response [23]. The poor bioavailability of constituents can be overcome by the development of nano-range drug vehicles or carriers, aimed at enhancing solubility, stability, and bioavailability, and showing a sustained-release effect in the wound area [24]. A dressing consisting of medicinal agents, especially with the enhanced application of nanotechnology could help in the rapid healing of full-thickness wounds, which can cause serious situations in the course of physicians healing them [25]. The traditional treatment of wounds fails to deliver a good therapeutic efficacy and has a low patient compliance. Frequent dosing or application of topical agents at the site of the wound, as well as frequent dressing, increases pain. For this reason, the approach of delivering medicinal agents in the form of a dressing reduces the dosing frequency and dressing intervals. Being the primary barrier to the external environment, the skin is continuously subjected to various injuries [26]. Skin wounds with a thickness less than 1 mm can regenerate themselves through the various healing processes, such as hemostasis, inflammation, proliferation, and restoration. However, chronic wounds, or in other words full-thickness wounds that extend towards the deeper dermis, tendons, and sometimes bones, entail severe treatment challenges, leading to permanent structural impairments [27]. Wounds categorized as non-life-threatening impact the quality of life, while some wounds allied to 5-year mortality predominate in many forms of cancer [28].

Thus, considering the need of patients, we developed a hydrogel dressing of a dendrimer-based therapeutic delivery of phytocompound-hesperidin, to improve the delivery of drug and enhance the process of wound healing. We evaluated the efficacy of dendrimer-mediated delivery of hesperidin both in vitro and in vivo.

## 2. Material and Methods

### 2.1. Materials

Hesperidin (Hsp) used for this study was obtained from Central Drug House (CDH). PAMAM dendrimer was received as a gift sample from Sigma Aldrich. Ketamine, and xylazine was obtained from M/s Myzen Enterprises Pvt. Ltd., New friend’s colony, Headquarter New Delhi, India. Chitosan powder (pharmaceutical grade with molecular weight 310–375 kDa) and sodium alginate were obtained from Sigma Aldrich, Taufkirchen, Germany. All other chemicals used in the study were of analytical grade.

### 2.2. Preparation of Hesperidin Loaded PAMAM Dendrimer (Hsp-PAMAM) Based Hydrogel Bandages

An adequate amount of PAMAM dendrimer was accurately weighed using a calibrated micropipette and subjected to evaporation under magnetic stirring for 24 h. After evaporation, Hsp (weight appropriately using weighing balance) in different concentrations (2.5, 5, 7.5 and 10 *w*/*v*%) was loaded into the PAMAM dendrimer. The solution was magnetically stirred for 24 h, subjected to evaporation and finally lyophilized, to obtain the dry powder of Hsp-PAMAM. For bandage preparation, sodium alginate was dissolved in deionized water and later mixed with the solution of chitosan, which was prepared by dissolving chitosan powder in deionized water containing 1% acetic acid at room temperature. Both solutions were mixed for about 30 min, with constant stirring [29]. The Hsp-PAMAM was then added to the solution of gels, stirred for about 15–20 min, homogenized, and lyophilized for 4–6 days, to obtain the hydrogel bandages.

### 2.3. Drug Loading and Entrapment Efficiency

To determine the drug loading and entrapment efficiency of hesperidin, Hsp-PAMAM was centrifuged at 3000 rpm for 15–20 min [30]. The supernatant was then discarded and re-dispersed in PBS (pH 7.4) solution. Absorbance was taken using a UV-visible spectrophotometer at 285 nm. The amount of drug loaded and entrapment efficiency were calculated by using formulas below:Drug loading capacity = [Amount of total entrapped drug/Total nanoparticle] × 100 
Entrapment efficiency = [(Weight of drug in total − Weight of free drug available)/Weight of drug in total] × 100

### 2.4. Characterization of Hesperidin PAMAM Dendrimer

#### 2.4.1. Fourier Transform Infrared Spectroscopy (FT-IR)

FT-IR of Hsp and Hsp-PAMAM were obtained using an FT-IR spectrophotometer (Perkin Elmer, Rodgau, Germany) in a range of 4000–400 cm^−1^_._ For each sample, potassium bromide pellets were made and analyzed in the above mentioned region.

#### 2.4.2. Differential Scanning Colorimetry (DSC)

Approximately 2–5 mg of Hsp and Hsp-PAMAM was sealed in a hermetically aluminum pan, and DSC thermograms were recorded at range of 30–400 °C using DSC 6 (Perkin Elmer, Rodgau, Germany) at a rate of 10 °C/min [31].

#### 2.4.3. Particle Size, Zeta Potential, and PDI Determination

The Hsp-PAMAM was subjected to estimation of the size of the particles. A Malvern zetasizer was used to determine the hydrodynamic diameter, surface potential, and polydispersity index (PDI).

#### 2.4.4. Morphology Examination (Transmission Electron Microscopy)

Surface characteristics of the prepared nano-formulation were determined using a transmission electron microscope (TEM). To determine the morphology, the sample was placed on a copper grid for a time interval of 10 min, followed by rinsing with water 2–3 times. The samples were then stained with 10 μL of 2% of a phospho-tungstic acid aqueous solution for 30 s. The stained grid was washed with distilled water, followed by air drying. The sample was scanned under a TEM (TECNAI G2 {200 kv} HR-TEM FEI, Amsterdam, The Netherlands).

### 2.5. Hemcompatibility Study

A hemolysis study for the prepared formulation was performed by the previously prescribed method with slight modifications. Approximately 3 mL of fresh human anticoagulated blood was taken, to obtain a 5% RBC suspension. To the RBC suspension, different concentration of Hsp-PAMAM were added (25, 50, and 100 μg/mL) and incubated at 37 °C for 30 min. PBS was taken as a negative control, whereas Triton-X served as a positive control. Triplets were prepared for each sample, which was then placed in an incubator at 37 °C for 30 min. The samples were subjected to centrifugation at 5000 rpm for 15–20 min and absorbances were noted at 545 nm using a UV-visible spectrophotometer. The hemolysis value was calculated using the formula [32]:Hemolysis percentage = [(Dt – Dn) / (Dpc – Dnc)] ×100
where Dt, Dpc, and Dnc represents the absorbance of the Hsp-PAMAM at various concentrations, the absorbance of positive control, and absorbance of the negative control. Microscopic techniques were then employed to examine the samples [33].

### 2.6. In Vitro Drug Release

An in vitro release study was carried out using a dialysis membrane procedure [34]. The membrane was washed under running water for around 4–5 h. It was then washed with 0.3% sodium sulfide solution for 1 min and kept in hot water at 60–70 °C for 3–4 min. The membrane was then subjected to acidification, by treating with 0.2% sulfuric acid for 2–3 min, washed with hot water at 60–70 °C, and kept overnight in PBS solution at pH 7.2. The formulation was then added to a dialysis bag and kept in a 100 mL beaker containing PBS pH 7.2. Samples were withdrawn at regular time intervals of 0, 1, 2, 3, 4, 5, 6, 8, 12, and 24 h, and the same amount of freshly prepared buffer solution was then added immediately, so as to maintain a sink condition [35]. Samples were analyzed using a UV-spectrophotometer at a specific wavelength of 285 nm.

### 2.7. Ex Vivo Drug Permeation Study

The visualization and depth of skin permeation of the prepared formulation were evaluated using confocal microscopy. Confocal microscope uses a laser beam of near IR region (830 nm), which was focused through an interconnected lens containing a beam splitter onto the skin region which was to be examined. The laser beam was then reflected by the rat skin and came back from the beam splitter and struck the detector. The confocal microscopy enabled in vivo imaging to a depth of 200–300 mm, with a lateral and vertical resolution of <1.25 μm and <5 μm [36].

### 2.8. Skin Permeation Enhancement Study

The skin permeation study of the prepared formulation on rat skin was analyzed using DSC and FTIR. On the top of excised skin, a hydrogel bandage was placed, which was compared with the untreated skin. Both the skin samples were placed on a Franz diffusion cell and left for 8 h. The samples were then washed with normal saline, cut into smaller sections, and dried in an oven; maintaining the temperature at 60 °C. The samples in dry form were then analyzed by DSC and FTIR, as per the previously published study [37].

### 2.9. In Vivo Wound Healing Activity

The animal experimental protocols were approved by the Institutional Animal Ethics Committee (IAEC), Jamia Hamdard, New Delhi, India under the CAHF reference number 1884. The 6–8-week-old Wistar rats were received from the central animal house facility of Jamia Hamdard, New Delhi. Standard conditions, such as adequate light and dark condition, stable room temperature, and adequate nourishment, were provided to the animals. After approval by the animal ethics committee of Jamia Hamdard and the committee for the purpose of control and supervision of experiments on animals (CPSCEA) Govt. of India, the animals were obtained [38].

A full-thickness wound was created, in order to evaluate and compare the wound healing ability of Hsp gel and various nano-preparations consisting of Hsp-PAMAM. All rats were anesthetized using a mixture of ketamine (75 mg/kg body weight) and xylazine (25 mg/kg body weight) by intraperitoneal route. An excisional full-thickness wound (1.5 by 1.5 cm^2^) was created on the dorsal side of the rat, posterior to the neck surface, after removing hairs with the help of depilator cream and using a scalpel blade. The whole body was cleaned and decontaminated with methanol. All the wounded rats were then placed in a propylene cage until the completion of the study [39]. The animals were categorized into six groups, containing six animals in each group, as shown in Table 1a.

Wounds were then treated with hesperidin gel and Hsp-PAMAM hydrogel (Hsp-P-Hyd) bandages (10%, 7.5%, 5% and 2.5%). Hsp gel was applied to the wound area using an adhesive bandage. With the use of a digital camera, the rate of wound contraction was evaluated, by observing the contraction of the wound area on 0 days, 7 days, and 14 days. Wound contraction was calculated:
Percent wound contraction = [(Initial size of wound size − Specific day wound size)/Initial size of the wound] × 100
Initial size of the wound

### 2.10. Histopathological Study

Four animals were terminated or euthanized after 14 days of treatment. The middle portion of the skin tissue was taken, transferred, and fixed immediately into 10% formalin solution (pH 7.2) [40]. Then, skin tissues were processed, submerge, or implanted into paraffin blocks and cut to 5 μm in thickness. The cut tissue sample was stained with Masson’s trichrome (MT) and hematoxylin and eosin (H&E), to obtain the histological slides. The slides were viewed under a light microscope (Olympus BX 50). Angiogenesis, epithelialization, and collagen development was assessed [41].

### 2.11. Skin Irritation Study

A skin irritation test of the prepared formulation was carried out in albino Wistar rats. Animals were stored under standard laboratory conditions in a light/dark cycle at room temperature. The prepared formulation was applied on the dorsal side of the rat, near to the posterior surface of the neck daily to the skin portion for about 7 days, to observe any kind of edema, erythema, or other kinds of reaction [42].

## 3. Results and Discussion

### 3.1. Preparation of Hesperidin Loaded PAMAM Dendrimer (Hsp-PAMAM) Based Hydrogel Bandages

A hesperidin-loaded PAMAM dendrimer was prepared appropriately, which showed the encapsulation efficiency and drug loading of 20% and 3.33%, respectively. The formulation was then incorporated into the chitosan-sodium alginate hydrogel, to form a bandage. The loading of Hesperidin into the PAMAM dendrimer was further confirmed by FTIR and DSC.

### 3.2. FT-IR

The FT-IR spectra of Hsp show characteristic absorption bands, due to various functional groups. Absorption bands at 3585.82, 3497.09, and 3399.68 cm^−1^ correspond to O-H stretching due to the presence of alcohol. The band at 3079.49 shows the presence of aromatic C-H stretching (alkane), while the absorption band at 2938.68 and 2850.91 cm^−1^ is attributed to CH_2_ stretching. The absorption bands at 1646.32, 1607.74, 1518.04, 1096.58, and 1070.54 cm^−1^ are attributed to C=O stretching, C=C stretching, C=C-C stretching, and last two are due to alkyl substituted C-O stretching. Methyl C-H bend (alkane) is obtained at 1446.67 cm^−1^. The observed peak was nearly the same as that reported in various literature data (Figure 1a). Our study is in correspondence with a previously published study [43].

The FT-IR spectra of the formulation showed absorption bands at 1878.75 and 1858.49 cm^−1^ from C-H bending (aromatic compound), while the band at 1818.95 showed C=O stretching. The bands at 1782.30, 1761.08, 1723.47, and 1707.08 cm^−1^ indicated C=O stretching. The band at 1627.03 cm^−1^ was due to C=C stretching (alkene), while 1569.16 and 1531.55 cm^−1^ are due N-O stretching. The band at 1446.67 cm^−1^ is due to methyl C-H bend (alkane). The band 1384.95 cm^−1^ is due to C-H bending (alkane), while 1350.23 and 1329.01 cm^−1^ both are due to O-H bending (alcohol). The band 1086.93 cm^−1^ is due to C-O stretching (alkyl substituted ether) (Figure 1b).

### 3.3. Differential Scanning Calorimetry (DSC)

Hesperidin is a flavonoid glycosides compound with an endothermic peak at 248.253 °C, which indicates the melting point of hesperidin, and this was authentic according to data available on hesperidin. The DSC thermogram of formulation showed an endothermic peak at 118.767 °C, which indicates that the drug had been incorporated (Figure 2a,b). A similar study was performed by another researcher, who revealed a decline in the peak after loading of hesperidin by the nanoparticle [44].

### 3.4. Particle Size, Zeta Potential, PDI Determination, and Transmission Electron Microscopy (TEM)

The particle size for the prepared nano-formulation obtained was 12.06 ± 3.292 nm (Figure 3a). The PDI (polydispersity index), which corresponds to the uniformity of particle size distribution, was 0.044. The zeta potential of the prepared nano-formulation was also determined using a Nano-ZS zeta sizer, and using zeta mode to determine the charge present on the surface of nano-formulation; it was found to be 12.48 mV. TEM imaging of the formulation showed a spherical morphology (Figure 3b). Therefore, the TEM analysis confirmed the morphology of the formulation.

### 3.5. Hemolysis Study

Compatibility of a formulation with RBC, especially for erythrocytes, is essential. The hemolysis effect of the prepared formulation at different concentrations of 25, 50, and 100 μg/mL were 0.33%, 0.47%, and 0.51%, respectively, which are within acceptable limits (Figure 4a,b). On the basis of the rules of the ASTM E2524-08 standard, a hemolytic value higher than 5% for the tested nanoparticles can cause hemolysis of the RBC. Therefore, the prepared formulation is hemocompatible, and hence it can be used within the body, but in lower concentrations. Figure 4c depicts the outcomes of the light microscopy images of the different concentrations of the prepared formulation. The hemolysis study showed that the shape or morphology of RBCs were not damaged by prepared the formulation. Thus, the hemolysis study on the RBCs suggested that the prepared formulation is biocompatible, and hence can be used safely within the body.

### 3.6. In Vitro Release Study

The percentage release profile of the Hsp-PAMAM hydrogel bandage was 86.367% release in 24 h (Figure 5). During the first 5 h, it showed a burst release of the drug from the formulation. After 5 h, the release of the drug from the formulation slowed down, which corresponded to the delayed release of the formulation. The accessibility of free hesperidin on the outer surface could be the reason for the burst release of the drug from the prepared formulation. After 12 h, it a showed sustained release of the drug, which might be due to affinity of the bonding drug and PAMAM dendrimer, which suggests that the release of the drug from the inner core of dendrimer was delayed after the burst release during the initial phase. The prepared formulation showed a sustained release pattern and can be used for delivery of the drug through a topical route. It is important to understand that the PAMAM dendrimer showed a strong bond with the guest cargo molecule, which confirmed the controlled release of drug at a higher pH [45].

### 3.7. Ex Vivo Permeation Study

To evaluate the permeation of Hsp-PAMAM and normal Hsp gel, a confocal microscopic study was performed. The untreated skin was kept as a control. The intensity of the red color represents the drug deposition in the rat skin. Rat skin treated with the control group and the drug showed maximum intensity at the skin surface of 0–5 μm, while the prepared formulation showed a deposition of drug in the epidermis up to 15–25 μm. The results, thus, show that the drug could efficiently be conserved in between the layers of the epidermis and dermis, which is essential for the therapy of a full-thickness wound (Figure 6). Our study is in accordance to a previously published study by Borowska, where they demonstrated the depth permeation of drug 8-methoxypsoralen using G3 and G4 PAMAM dendrimers. The formulation was similarly placed on rat skin using Franz diffusion cells, and the extent of permeation was evaluated through a confocal microscopic study. The results suggested a feasible effect of both dendrimers in transdermal delivery, which could improve the safety and effectiveness of PUVA (Psolaren+ UVA light) therapy [46]. This experiment was carried out to evaluate the extent of penetration of hesperidin, after being loaded into the core of a PAMAM dendrimer.

### 3.8. Skin Permeation Enhancement Study

To evaluate the permeation of Hsp-PAMAM from the hydrogel bandage into the skin and its mechanism of permeation, FTIR and DSC were used. In the FTIR of normal skin (Figure 7a), the absorption band from 3000 to 2700 cm^−1^ occurred due to the C-H stretching motion of the alkyl group present in both proteins and lipids. Absorption bands 2926.14 and 2853.81 cm^−1^ were due to asymmetric and symmetric C-H stretching in the lipids. Bands at 1639.56 and 1550.83 cm^−1^ were due to amide I and amide II stretching vibrations of the proteins. Amide I bands arise from C=O stretching vibration, and amide II band from C-N bending vibration. The amide I band consists of component bands that represent the various structures of keratin. The FTIR spectra of the treated skin showed absorption bands at 3576.18, 3421.87, and 3062.13 cm^−1^ (Figure 7b). The band at 2855.73 cm^−1^ represents C-H stretching (alkane), 1750.48 cm^−1^ band represents C=O stretching, and 1665.60 cm^−1^ band represents C=C stretching (alkene). The absorption band at 1462.11 cm^−1^ shows C-H bending (alkane), the band at 1426.42 cm^−1^ O-H bending, and 971.20 cm^−1^ represents C=C bending (alkene). Both spectra of hesperidin and rat skin treated with formulation showed similar characteristic peaks, with minute differences. This difference in peaks can be attributed to the chemical interaction between the skin and Hsp-PAMAM.

The DSC thermogram of normal rat skin was compared with rat skin treated with the prepared formulation. The DSC thermogram of the normal rat skin showed an endothermic peak at 146.386 °C, which was due to the denaturation of protein, while the DSC thermogram of the rat skin treated with the formulation showed a sharp endothermic peak at 116.227 °C (Figure 7c,d). The shift in peak to a lower value indicates the interference in the junction of the stratum corneum, which was required for permeation of the drug.

### 3.9. In Vivo Wound Healing Activity

The efficiency of the Hsp-P-hyd was determined using a full-thickness wound created on rats. There is extensive evidence of the safety and effects of hesperidin, such as being anti-radical, anti-oxidant, antibacterial, and anti-inflammatory; with its main effect being prompting faster wound healing [47,48,49]. Previous studies have also suggested the potential of Hsp in early collagen synthesis and deposition, early epithelialization, and increased cellular proliferation [50].

Figure 4 shows images of full-thickness wounds taken from day 0 to day 14 for the control group, plain hesperidin gel, and Hsp-PAMAM (2.5%, 5%, 7.5%, and 10%). Almost every group showed wound contraction over time, while the hesperidin gel showed accelerated wound closure in comparison to the control group (Figure 8a,b). It was found that the control group had inflammation and infection, with incomplete wound closure. Hsp-P-Hyd 10% exhibited the fastest wound closure, which was due to the properties of the hesperidin reaching deeper into the dermis with the help of the PAMAM dendrimer [51]. No sign of inflammation or infection, and complete wound closure, was observed. To assess the process of wound healing, the wound closure percentage was determined (Figure 8a, Table 1b).

Percent of wound closure for the control group after 7 and 14 days was 24.665 ± 0.94 and 79 ± 1.41, whereas for hesperidin gel it was found to be 79 ± 1.41 and 79 ± 1.41. The best was obtained for 10% formulation, where the wound closure was 49.105 ± 0.62 and 98.9 ± 0.42.

### 3.10. Histopathological Study

The histologic sections of the control group (untreated) and formulation treated group are shown in Figure 9a,b. Histopathological evaluation of the control group showed degenerated neutrophil and eosinophils infiltration. Hesperidin gel showed the formation of the epidermal layer at the injured site. The wound was completely covered by an epidermal and granular layer. Hsp-P-Hyd 10% produced the maximum contraction of the wound, formation of epidermal layer, and remodeling. The epidermal layer was totally formed, due to the healing property of hesperidin (Figure 9a). In the case of MT (Masson’s trichrome) staining, this showed that among all the prepared formulations, Hsp-P-Hyd 10% revealed the most improved collagen synthesis during the process of wound healing. On the contrary, the collagen formation rate and deposition were very low in the control group (Figure 9b). Therefore, it can be concluded that 10% hesperidin could help in increasing the collagen synthesis rate [52]. The wound healing effect of hesperidin in diabetes-induced rats has also shown potential effects, by accelerating vasculogenesis and angiogenesis via upregulation of TGF-β, VEGF-c, Smad-2/3 mRNA, and Ang-1/Tie-2 expression, which in congruence suggests promising effects for treatment of full-thickness wounds [53].

### 3.11. Skin Irritation Study

Hesperidin is a nonirritant, as it is a phytoconstituent, but the prepared formulation contained various chemical constituents, so it becomes necessary to conduct a skin irritation study (Figure 9c). The rats were observed for 7 days, to observe any toxic reaction or harmful effect. The score of this study was 0.14 for the formulation, which suggests no signs of dropsy or erythema on the skin. Values between 0 to 9 suggest non-irritant potential, hence the prepared formulation is safe and compatible for delivery via the topical route (Table 1c).

## 4. Conclusions

In this study, different concentrations of hesperidin were loaded into a dendrimer, to increase the wound healing rate, and their efficacy was evaluated in a rat model. The obtained results showed that Hsp-P-Hyd 10% had a better wound healing effect compared to the other groups in the study. The hemolysis study suggested its biocompatibility with RBCs and that it can be used for the treatment of full-thickness wounds. The in vitro study revealed that 10% of the prepared formulation (hesperidin) had a better efficiency for treating the wound compared to the control or other prepared formulations of hesperidin. Around 98.9% of the wound was healed after 14 days of the animal study. Based on the obtained results, it can be concluded that 10% of hesperidin formulation is promising for the successful treatment of skin-related injuries or wounds.

## Figures and Tables

**Figure 1 biosensors-12-00462-f001:**
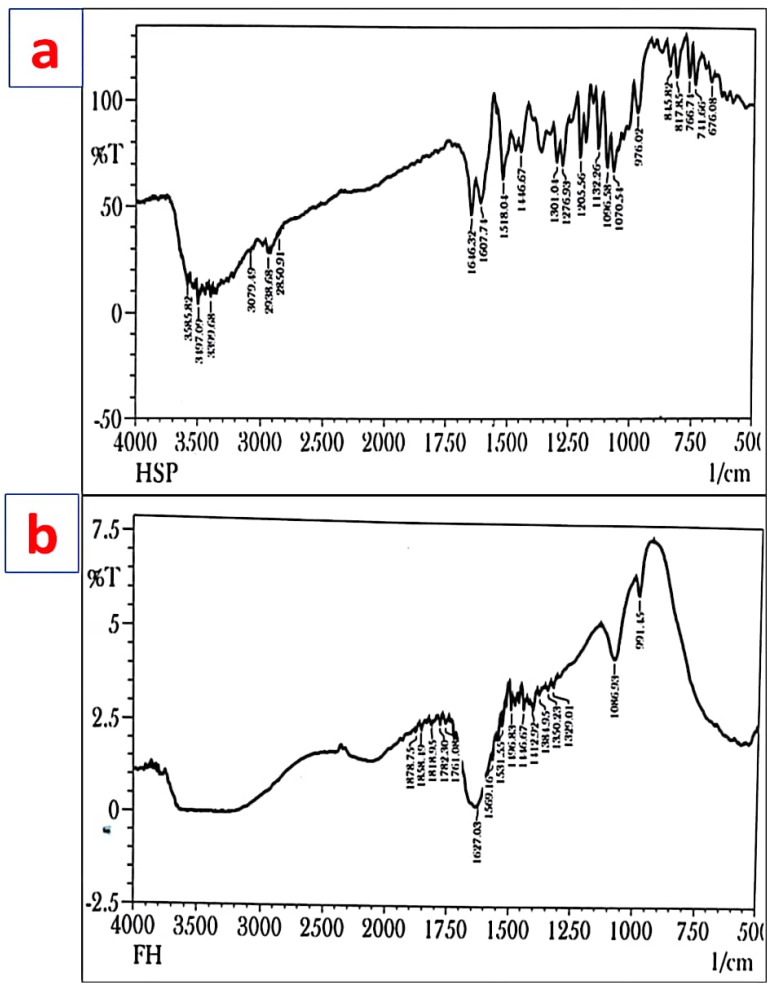
FTIR spectra of (**a**) Hesperidin (Hsp) and (**b**) Hsp-PAMAM.

**Figure 2 biosensors-12-00462-f002:**
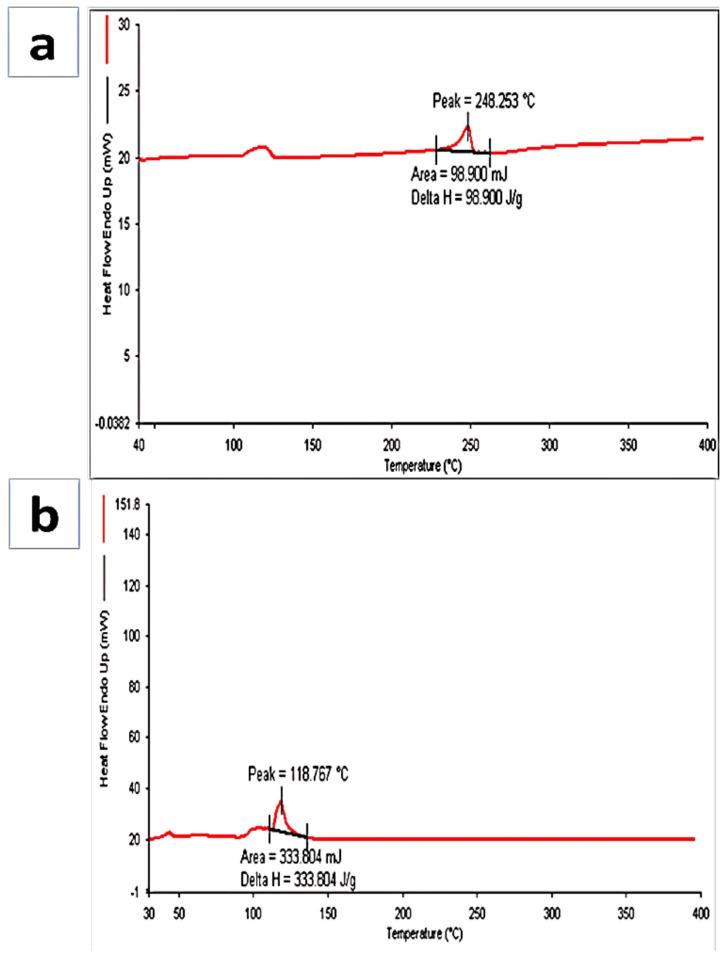
Representation of DSC analysis of (**a**) Hsp and (**b**) Hsp-PAMAM.

**Figure 3 biosensors-12-00462-f003:**
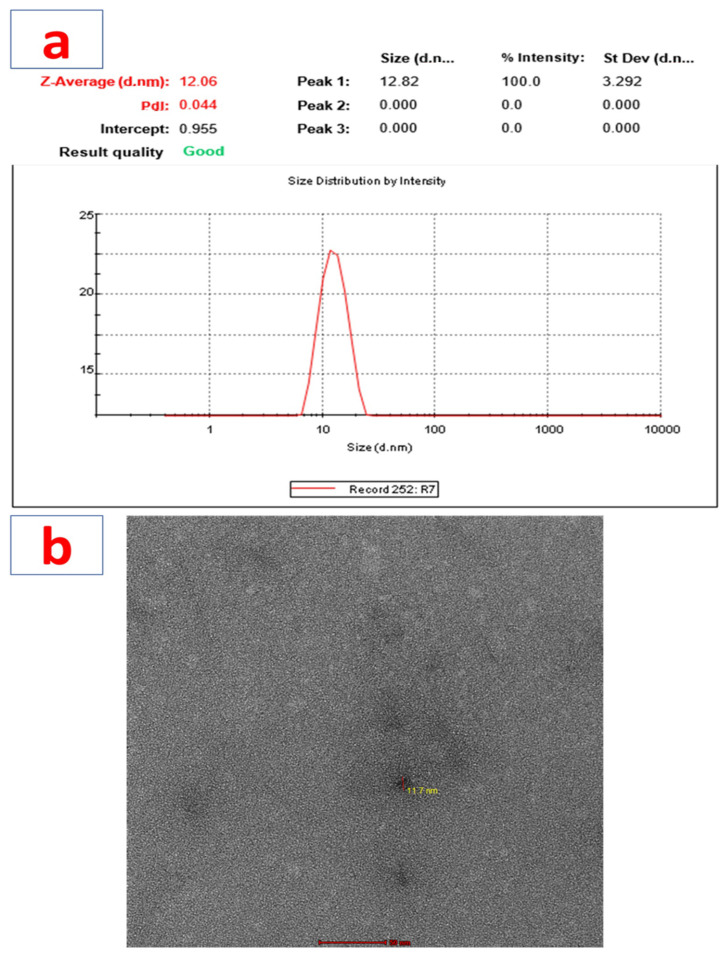
(**a**) Particle size determination using a Malvern Zetasizer and (**b**) particle morphology with TEM.

**Figure 4 biosensors-12-00462-f004:**
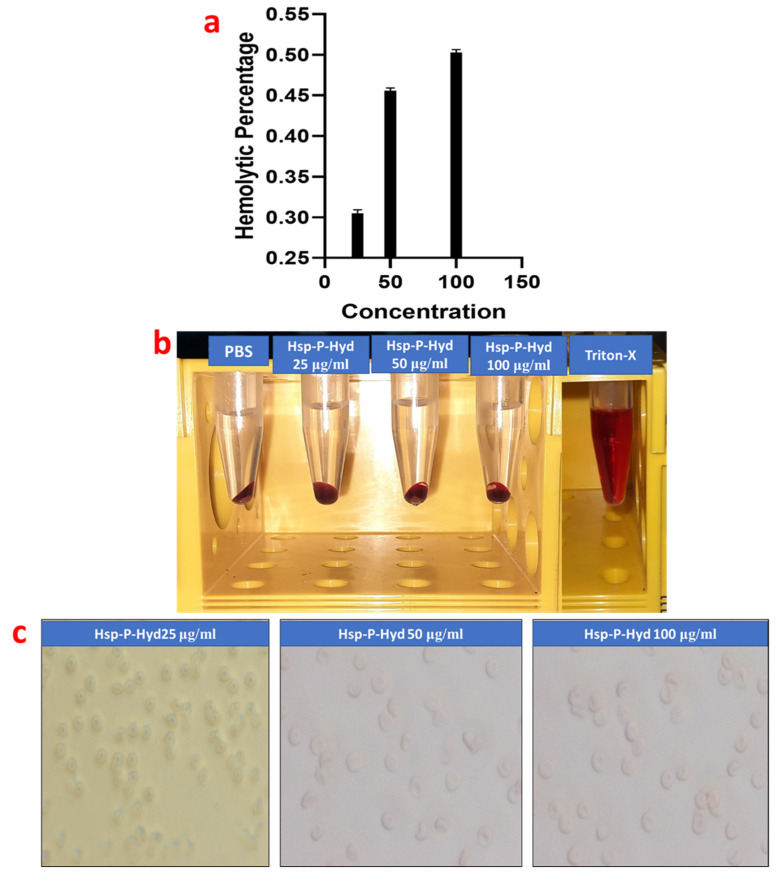
Representation of percentage hemolysis of (**a**) Hsp-PAMAM at concentration 25 μg/mL, 50 μg/mL, and 100 μg/mL, (**b**) Representation of hemolysis using different concentrations of Hsp-PAMAM (left to right: Treated with Hsp-PAMAM at concentration 25 μg/mL, 50 μg/mL, and 100 μg/mL, PBS and positive control), (**c**) Optical microscopic images of RBC treated with (i) 25 μg/mL, (ii) 50 μg/mL, and (iii) 100 μg/mL of Hsp-PAMAM.

**Figure 5 biosensors-12-00462-f005:**
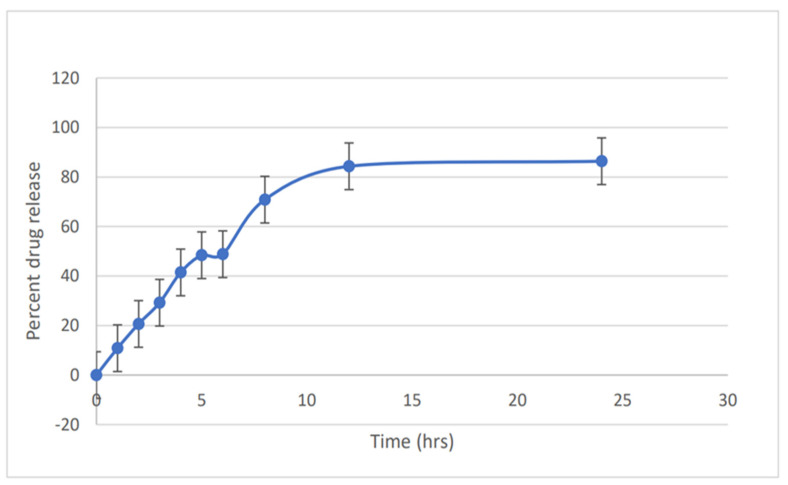
Percentage release study of Hsp-PAMAM.

**Figure 6 biosensors-12-00462-f006:**
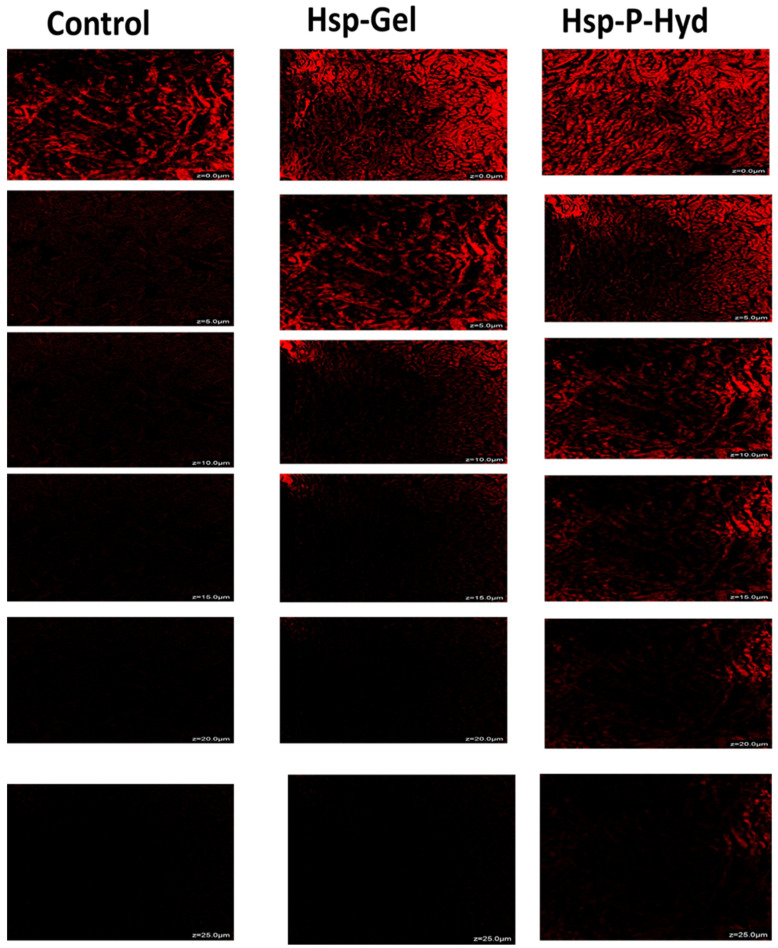
Confocal microscopic images of skin treated with a control, plain Hsp gel, and Hsp-PAMAM hydrogel bandage. In the control and Hsp gel-treated groups, the intensity of red fluorescence reached only 5 µm, while a higher penetration was observed in those treated with Hsp-PAMAM, reaching a depth of 15 to 20 µm.

**Figure 7 biosensors-12-00462-f007:**
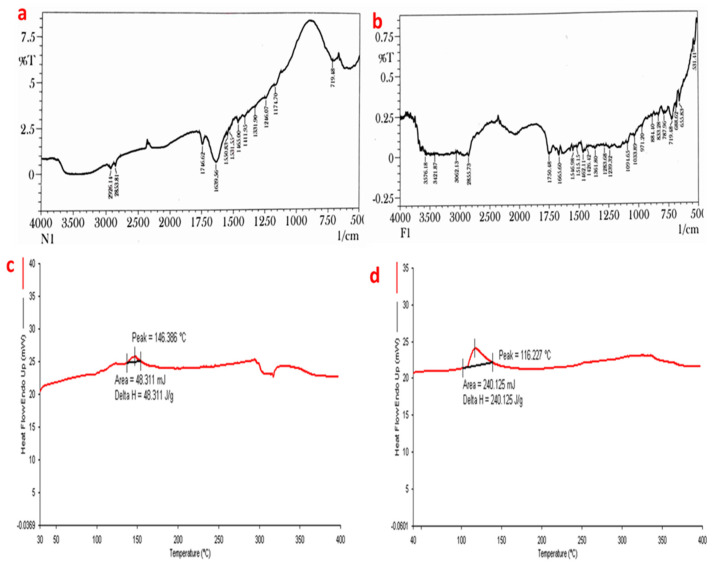
Skin permeation enhancement study evaluated using FTIR and DSC: (**a**) FTIR of normal skin and (**b**) Hsp-PAMAM-treated skin. (**c**) DSC of normal skin and (**d**) Hsp-PAMAM treated skin.

**Figure 8 biosensors-12-00462-f008:**
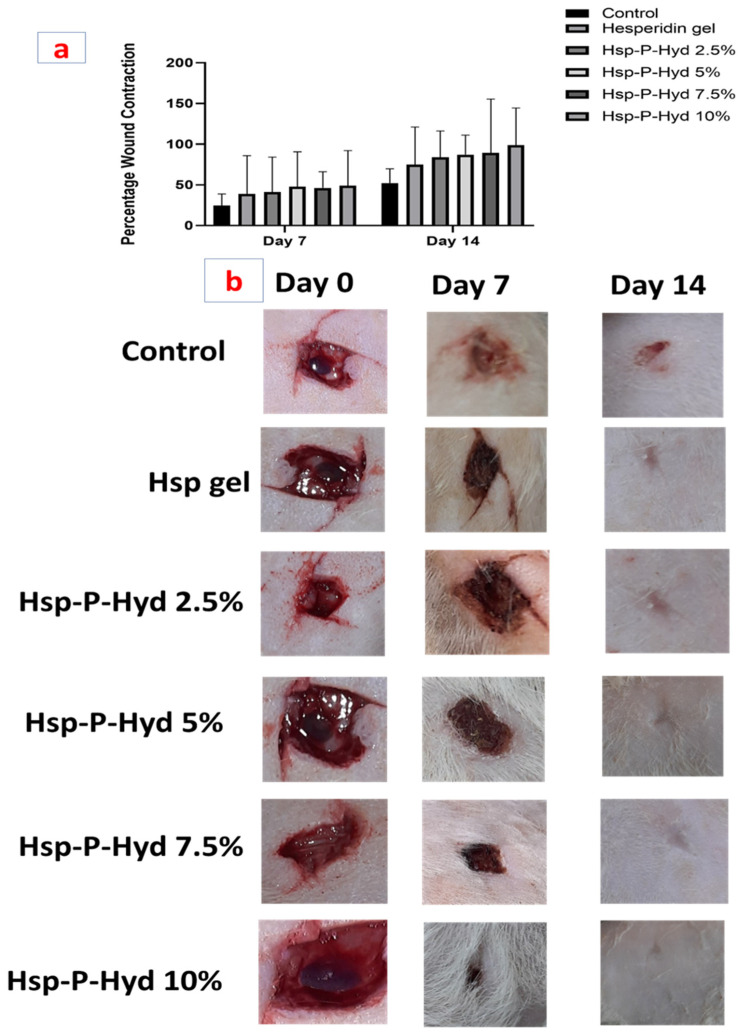
(**a**) Representation of percentage of wound contraction at 7 and 14 days (**b**) Representation of wound healing progression from 0 day to 14 day.

**Figure 9 biosensors-12-00462-f009:**
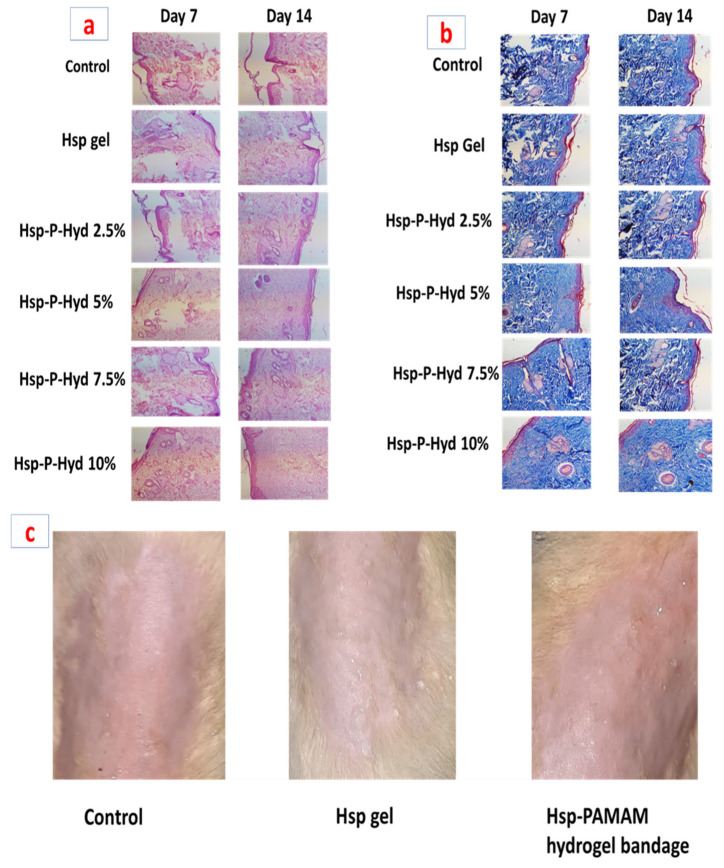
Representation of Histopathological study (**a**) H&E staining of wounds at 7-day and 14-day. (**b**) MT staining of wounds at 7-day and 14-day. (**c**) Representation of a skin irritation test after application of Hsp gel and Hsp-PAMAM hydrogel bandages.

**Table 1 biosensors-12-00462-t001:** (**a**) Distribution of animals for the in vivo wound healing study; (**b**) Percentage wound contraction evaluation post treatment with Hesperidin gel, Hsp-P-Hyd 2.5%, Hsp-P-Hyd 5%, Hsp-P-Hyd 7.5%, Hsp-P-Hyd 10%. Wounds not treated were considered as control; (**c**) Skin irritation evaluation after treatment with Hsp gel and Hsp-P-Hyd 10%.

(a): Distribution of Animals for In Vivo Wound Healing Study								
**Groups**	**Treatment**		**No of Animals Required/Group**			**Route**		
1	Control		6			Topical		
2	Hesperidin gel		6			Topical		
3	Hsp-P-Hyd—10%		6			Topical		
4	Hsp-P-Hyd—7.5%		6			Topical		
5	Hsp-P-Hyd—5%		6			Topical		
6	Hsp-P-Hyd—2.5%		6			Topical		
**Total number of animals = 36**								
(b): Percentage wound contraction evaluation post treatment with Hesperidin gel, Hsp-P-Hyd 2.5%, Hsp-P-Hyd 5%, Hsp-P-Hyd 7.5%, Hsp-P-Hyd 10%. Wounds not treated were considered as control								
**Groups**			**Day 7**			**Day 14**		
Control			24.665 ± 0.94			52 ± 1.41		
Hesperidin gel			38.995 ± 0.47			74.998 ± 0.46		
Hsp-P-Hyd 2.5%			41 ± 0.48			83.93 ± 0.098		
Hsp-P-Hyd 5%			47.665 ± 0.47			87.263 ± 0.85		
Hsp-P-Hyd 7.5%			46.33 ± 0.46			89.16 ± 0.14		
Hsp-P-Hyd 10%			49.105 ± 0.62			98.9 ± 0.42		
(c): Skin irritation evaluation after treatment with Hsp gel and Hsp-P-Hyd 10%.								
**Rat group**	**Score after days**							**Mean score**
	1	2	3	4	5	6	7	
Control	0	0	0	0	0	0	0	0
Hesperidin gel	0	0	0	0	0	0	0	0
Hsp-P-Hyd 10%	0	0	0	1	0	0	0	0.14

## Data Availability

Not applicable.

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
