# Peer review of "Amelioration of Full-Thickness Wound Using Hesperidin Loaded Dendrimer-Based Hydrogel Bandages"

_biosensors, 2022, doi:10.3390/bios12070462_

Round 1

Reviewer 1 Report

The article is dedicated to the study of amelioration of full thickness wound using Hesperidin loaded dendrimer-based hydrogel bandages. The subject matter of the manuscript addresses a very interesting problem related to wound healing. The authors proposed the development of a dendrimer-based hydrogel bandage to ameliorate full thickness wounds using Hesperidin as an active agent.

The manuscript is well-written and keeps editorial standards. The material is presented in a clear way. The introduction provides scientific background and all necessary information about the material; motivation seems to be fully justified. The experimental methodology is sufficient.

I recommend accepting the article after a few minor corrections, as listed below:

1. Please correct typos (for example, in line 17);

2. The descriptions in Figure 1 should be corrected so that the peak designations can be read.

3. The font in Figure 2a should be larger. In particular, the scale and description on the graph in Figure 2a should be clearly visible;

4. The descriptions in Figure 3 should be corrected.

5. The font in Figure 4 should be larger. 

Reviewer 2 Report

Questions/recommendations for the authors:

1. Page 2, line 60. please explain PANAM.

2. Page 2, line 97. please explain CDH, line 98 SIR

3. Section materials and methods. It must be expanded and added more details e.g. how were the samples prepared for the measurement, which instrument was used etc. mainly the methods FT-IR, differential scanning colorimetry and particle size, zeta potential and PDI (it is necessary explain the abbreviation)

4. In vitro and in vivo is usual to write in italics.

5. in chapter 2.7 is used skin in the experiment but it is not clear the source of the skin e.g. animals (rat, mouse...) or human (healthy volunteers). In chapter 2.8 is used rat skin but from the description is unclear that was the skin after application of hydrogel?

6. The page 5, line 201-2022. Is it good idea use the methanol for disinfection? Is very toxic and could influence the further experiments.

7. Skin irritation evaluation is described without any explanation. You have to add more details.

8. Page 6, line 240 is missing space between the numbers.

9. Fig 1. have to be remade. Wrong name on B, some of the par of the pictures is on the line and etc.

10. Fig.2 particle morphology b) and optical microscopy images e). These images have to be improved. The visibility is not very good. Generally, almost images have to be improved.

11. Fig 4. C) the squares are definitely black. No differences between the pictures. Is it correct?

12. Page 9, line 331 is written that the animals were treated by plain hesperidin gel and Hsp-PANAM. Do you use also PANAM alone for the wound treatment?

Reviewer 3 Report

The manuscript reports a study where the authors claim to have prepared hesperidin-loaded dendrimer bandages with visible improvements in wound healing. Unfortunately, the manuscript is quite poorly written, and at certain places way beyond grasping. Most of the figures in the PDF file were not even legible, and I could not understand the presented data. With such low figure resolution, it becomes impossible for me as a reviewer to comment on the data. I am confused why the authors had to compress all their data into mere four figures whereas the data should be spread across at least eight figures with much higher resolution. PAMAM is not a standard abbreviation and should be written in its full form when mentioned for the first time. The dendrimers were a gift from "SIR" (line 98). Is it a person or a chemical supplier? Were these dendrimers characterized before loading?

From the data, or whatever I could understand from the text as the figures were beyond understanding, I am not at all convinced that hesperidin was encapsulated in these dendrimers. It was indeed present within the mix but it does not necessarily mean encapsulation. In Figure 3b, the TEM image is hazy and I am not sure what is shown there. There is a red stripe at the bottom margin which I guess is the scale bar although I have no idea what it means. No mention of the scale bar could also be noted in the figure legend also. In Figure 3c, the word "percentage" in the y-axis legend is wrongly spelled. The concentration of 25 ug/ml was not written on the x-axis whereas 150 ug/ml was mentioned which has got nothing to do with the paper. The spelling of hemolysis keeps oscillating between American (hemolysis) and British (haemolysis) forms throughout the manuscript. In Figure 3c, it is in British form when used as a y-axis legend but in American form when mentioned in the same figure legend. In the in vivo study, six animals were chosen for each group making it 36 animals for the entire study. There is no mention of statistical power calculation which makes me question this rather arbitrary number of six and how the authors got to this number? In summary, I cannot recommend acceptance of this manuscript.

Round 2

Reviewer 2 Report

2.7. and 3.7 Ex vivo drug permeation study. For me is still unclear Fig.6. One column of pictures is control, second plain Hsp-gel and the last one Hsp-PANAM hydrogel bandage. What are the differences between the lines?

Some of the figures are still unclear with the wrong visibility. Some of the methods and results is not sufficient describe

To comments 6 is given the article to explain the using of methanol as the disinfection. In this article, the methanol is not mentioned.  

Reviewer 3 Report

The manuscript has improved considerably in its revised form and can be accepted after minor textual edits. 
